# The Effects of Pre-Existing Antibodies on Live-Attenuated Viral Vaccines

**DOI:** 10.3390/v12050520

**Published:** 2020-05-08

**Authors:** Darren Z. L. Mok, Kuan Rong Chan

**Affiliations:** Program in Emerging Infectious Diseases, Duke-NUS Medical School, Singapore 169857, Singapore; darren.zl.mok@u.duke.nus.edu

**Keywords:** live-attenuated vaccine, vaccine immunogenicity, antibody-dependent enhancement, vaccine immune interference, pre-existing antibodies

## Abstract

Live-attenuated vaccines (LAVs) have achieved remarkable successes in controlling virus spread, as well as for other applications such as cancer immunotherapy. However, with rapid increases in international travel, globalization, geographic spread of viral vectors, and widespread use of vaccines, there is an increasing need to consider how pre-exposure to viruses which share similar antigenic regions can impact vaccine efficacy. Pre-existing antibodies, derived from either from maternal–fetal transmission, or by previous infection or vaccination, have been demonstrated to interfere with vaccine immunogenicity of measles, adenovirus, and influenza LAVs. Immune interference of LAVs can be caused by the formation of virus–antibody complexes that neutralize virus infection in antigen-presenting cells, or by the cross-linking of the B-cell receptor with the inhibitory receptor, FcγRIIB. On the other hand, pre-existing antibodies can augment flaviviral LAV efficacy such as that of dengue and yellow fever virus, especially when pre-existing antibodies are present at sub-neutralizing levels. The increased vaccine immunogenicity can be facilitated by antibody-dependent enhancement of virus infection, enhancing virus uptake in antigen-presenting cells, and robust induction of innate immune responses that promote vaccine immunogenicity. This review examines the literature on this topic and examines the circumstances where pre-existing antibodies can inhibit or enhance LAV efficacy. A better knowledge of the underlying mechanisms involved could allow us to better manage immunization in seropositive individuals and even identify possibilities that could allow us to exploit pre-existing antibodies to boost vaccine-induced responses for improved vaccine efficacy.

## 1. Introduction

“It’s time to close the book on infectious diseases, declare the war against pestilence won, and shift national resources to such chronic problems as cancer and heart disease” [1]. Contrary to this infamous statement, long misattributed to the former US Surgeon General Dr. William H. Stewart, and despite advances in healthcare and technology, we remain extremely vulnerable to the threat of communicable diseases. In the last ten years alone, we have experienced the pandemic spread of swine-origin H1N1 influenza, the West African Ebola epidemic, the resurgence of yellow fever, the Zika virus emergency, and the return of a global coronavirus threat [2,3,4,5,6]. With the continued emergence and re-emergence of new and current viral pathogens, it is imperative that we continue to design new strategies to combat their spread and prevent human disease.

Among the various methods to impede viral transmission, vaccines are widely heralded as one of the most effective medical interventions. Indeed, since the pivotal discovery by Edward Jenner over two hundred years ago, vaccination has seen tremendous success in reducing the burden of viral diseases such as polio, yellow fever, measles, mumps, rubella, hepatitis A, hepatitis B, and influenza [7,8]. However, perhaps the greatest testament to their success is the eradication of smallpox in 1980 [9]. According to the World Health Organization (WHO), vaccination now saves over 2.5 million lives annually and prevents even more cases of illnesses and serious disabilities [10]. Moreover, taking into account both treatment costs and lost productivity due to death and disability, vaccines are estimated to provide billions of dollars’ worth in cost savings between 2011 and 2020 [11]. In addition, at high enough coverage, vaccination can impart both individual and community protection in the form of herd immunity, safeguarding those who are legitimately incapable of receiving immunization. On the other hand, reduced vaccine coverage threatens to cause a resurgence of vaccine-preventable diseases, including measles, mumps, and pertussis [12]. With such accolades, vaccination is widely considered as the crowning achievement of public health, and the WHO touts it as the most cost-effective method to prevent infectious diseases. Given their remarkable track records, vaccination is likely to remain a cornerstone of antiviral strategies, especially with the increasing prevalence of epidemic viral diseases.

The last few decades have witnessed remarkable advances in vaccine design and development, with new and innovative technologies redefining our approaches to vaccine production. Nevertheless, one particular class of vaccines has withstood the test of time and has remained in use ever since the advent of immunization: the live-attenuated vaccines (LAVs). Comprised of living but attenuated microorganisms, this group, which also includes replication-competent viral vectors, retains the capacity to replicate in vivo but does not cause disease in humans. Due to their ability to mimic a natural infection and activate innate immune responses, LAVs are able to induce long-lasting, robust cellular and humoral immune responses without the need for adjuvants [13]. This provides an added advantage over their inactivated counterparts, which often require adjuvants to stimulate innate immune responses for the induction of adaptive immune responses. Active viral replication within cells enables antigen processing via major histocompatibility complex (MHC) class I, which is pivotal in activating cytotoxic CD8+ T-cells that facilitate clearance of virus-infected cells [14]. Furthermore, LAVs contain a repertoire of antigens similar to the wild-type organism, allowing them to present all epitopes in their native conformation to antigen-presenting cells [15]. Given their potential in activating robust CD8+ and CD4+ responses, polio, recombinant vesicular stomatitis virus (VSV), recombinant adenovirus, and attenuated measles vaccines have also been proposed as potential viral vectors for the delivery of tumor antigens [16,17].

One of the greatest challenges for the development of LAVs is to ensure that the vaccine is safe and does not cause serious adverse events. Moreover, they should not replicate in vectors to prevent viral transmission, and caution must be taken to ensure that the attenuated viral strains do not revert to a wild-type virus. Therefore, to ensure the safety of LAVs, there is usually a tendency to select LAV candidates with limited replicative potential in order to reduce the risk of adverse events. However, it is also critical to ensure that LAV candidates are not over-attenuated so as to induce sufficient innate and adaptive responses necessary for vaccine efficacy. Vaccine immunogenicity and efficacy, however, are difficult to assess in animal models and may not correlate with clinical efficacy. An example is the Dengvaxia^®^ vaccine manufactured by Sanofi Pasteur, which generated protective antibody and T-cell responses against all dengue virus serotypes in non-human primates, but in phase II and III clinical trials, only low to modest efficacy was observed against DENV serotype 1 and 2 [18,19]. Controlled human infection model (CHIM) trials are now considered for evaluation of vaccine candidates but the ethical, laboratory, scientific, and governance issues should be carefully managed [20].

With an anticipated increase in the demand for vaccines to control viral epidemics and high-endemicity of virus infection worldwide, the presence of pre-existing antibodies caused by infection or vaccination with an antigenically similar virus may influence the outcome of LAV immunogenicity and efficacy. Thus, vaccines that are administered to the adult population will have to consider the role of pre-existing immunity on vaccine efficacy. Conversely, it is critical to determine whether the administration of LAV generates sufficient antibody responses that protect against wild-type virus infections and not cause worse disease outcomes. For vaccines that are administered to neonates, the presence of passively acquired maternal antibodies within the first six months may also interfere with vaccination [21,22]. The potential effects of pre-existing antibodies on LAV and virally vectored antigens will hence be the focus of this review. In this review, we examine the scenarios in which pre-existing antibodies can either enhance or inhibit LAV efficacy, as well as the underlying mechanisms involved. A better understanding will allow us to tailor our vaccination schedules or vaccine doses, to ensure that LAV efficacy will not be compromised by the presence of pre-existing immunity.

## 2. Effects of Pre-Existing Immunity on Live Vaccines

### 2.1. Measles

Prior to the advent of vaccination, the incidence rate of measles was so high that infection by the measles virus was basically considered an inevitability [23,24]. The virus, which is spread by the respiratory route, is highly contagious and infects over 90% of individuals by age 18 in the pre-vaccination era [24,25]. About seven to eight million children were estimated to have died from measles infection each year during this period, with many others suffering from disease complications [26]. However, in 1936, society experienced a turning point in the fight against measles with the development of the first live-attenuated measles vaccine (MV) [27].

The live-attenuated MV is one example of a highly successful LAV, and its introduction has transformed measles from a complicated disease into a triviality in most developed countries, although measles still forms a significant disease burden in developing nations [28,29,30]. The wild-type virus was first isolated in 1954 by Dr. John F. Enders and his team from the blood of an 11-year-old boy named David Edmonston, who became the namesake of the viral strain that would eventually become the first measles vaccine [31]. Serial passage of the wild-type Edmonston strain in human and chicken embryo fibroblast tissue culture resulted in a virus with reduced virulence. However, the ability of the parental strain to induce protective immunity is retained. The MV is highly immunogenic, inducing both humoral and cellular immunity at magnitudes comparable to that of a natural infection, although antibody titers induced are often lower [32,33]. Investigators have demonstrated that this protection is highly robust, and could last for as long as 20 years after vaccine administration [34]. Its excellent safety profile, highly immunogenic nature, and low possibility of reversion to virulence has also placed it as a promising candidate for use as a viral vector to deliver heterologous antigens [35].

Some of the first evidence that describes the role of pre-existing immunity on live vaccines comes from the measles vaccine. The efficacy of live MV was found to be often hampered by pre-existing immunity at the time of vaccination, and this effect is best illustrated in infants who are born to measles-immune mothers. During gestation, infants acquire protective antibodies as a result of transplacental transfer of maternal IgG antibodies [36]. These antibodies, while protective against infection, can also suppress infant responses to immunization. Indeed, studies have shown that vaccinating infants born to measles-immune mothers before or at the age of six months often results in seroconversion failure [37,38]. By contrast, immunization campaigns with MV between 9 to 12 months of age are relatively successful [39,40]. This is likely explained by waning maternal antibody levels over the period of 6 to 12 months, where antibody titers fall below the inhibitory threshold required for successful vaccination [41]. Moreover, the claims that pre-existing antibody titers can impact MV efficacy are further supported by animal studies. For instance, following the intravenous transfer of varying titers of MV neutralizing antibodies, immunization of mice with a recombinant measles vaccine (rMV) expressing Simian Immunodeficiency Virus (SIV) gag protein was significantly inhibited when pre-existing antibody titers were above 500mIU/mL of serum [35]. Likewise, investigations in cynomolgus macaques by van Binnendijk et al. revealed that pre-existing antibody titers as low as 0.1IU/mL abrogated the development of antibodies following vaccination with MV or a recombinant vaccinia virus vector expressing measles antigens [42]. Interestingly, while antibody induction by MV is negatively impacted by pre-existing immunity, the effect on cellular responses seem unaffected. For example, infants vaccinated with MV at age 6 or 9 months followed by the measles-mumps-rubella (MMR) vaccine at 12 months generated equivalent T-cell responses to control infants given only MMR at age 12 months [43]. Thus, experimental models and clinical data support that the inhibition of MV vaccination in neonates is more likely due to interference from maternal antibodies rather than the immaturity of the neonatal system.

### 2.2. Adenovirus and Adeno-Associated Virus

Adenoviruses (AdV) and adeno-associated viruses have been widely studied as a potential viral vector for cancer gene therapy and infectious diseases. This double-stranded DNA virus holds several advantages as a vector, including but not limited to the ability to induce robust cellular and humoral responses, high expression of transgenes, favorable safety profiles, and no risk of integration into the host genome. The most commonly used adenoviral vector is AdV serotype 5 (Ad5) and has been tested in more than 400 gene therapy trials. However, the large majority of the human population having pre-existing immunity to AdV limits its widespread use in the clinics. Indeed, a recent international cross-sectional serological survey demonstrated that greater than 80% of the study participants possessed neutralizing antibodies against Ad5 [44]. In a similar line of investigation, Mast et al. found that 85.2% of their study participants from the US, Europe, Thailand, Africa, and Brazil were seropositive for anti-Ad5 neutralizing antibodies [45]. These pre-existing antibodies have been shown to neutralize the Ad5 vectors after administration, thereby lowering their efficacy and transgene expression. The most convincing evidence is from the large-scale clinical trial (STEP) that tested an Ad5-based HIV-1 vaccine, where reduced efficacy of Ad5 was found to be associated with subjects who had pre-existing immunity to Ad5 [46]. In support of this theory, the clinical trial for the Ad5-based Ebola vaccine showed that the low-dose vaccine of 4.0 × 10^10^ viral particles was weakened by pre-existing immunity, whereas immunogenicity was enhanced at a higher dose of 8.0 × 10^10^- 1.6 × 10^11^ viral particles. In a mouse model, pre-existing anti-Ad5 neutralizing antibodies were observed to severely hamper the induction of both cellular and humoral responses by a recombinant Ad5-Ebolavirus glycoprotein vaccine [47]. Likewise, a recombinant Ad5 vector expressing the Human Immunodeficiency Virus-1 (HIV) gag gene showed diminished, but not complete, abrogation of gag-specific cellular immune responses following the vaccination of rhesus monkeys pre-exposed to an empty Ad5 vector [48]. Nonetheless, there exists conflicting data that pre-existing anti-Ad5 may not affect the induction of cytotoxic T-cell responses, indicating that more studies may be required to resolve these discrepancies [49].

The high seroprevalence of anti-Ad5 immunity has pivoted the development of genetically-modified AdV and the search of rarer AdV serotypes for use as vectors, in hopes that these viruses may circumvent the effects of pre-existing anti-Ad5 immunity [50,51]. However, there remains the possibility of cross-reactivity between antibodies against the different AdV serotypes, which could in turn potentially influence the efficacy of these rarer AdV as vaccine vectors. Indeed, investigations by Heemskerk et al. have shown that Ad5-specific CD4^+^ T-cells could cross-react with other AdV serotypes including but not limited to Ad1, Ad3, Ad7, and Ad35, suggesting that the different AdV serotypes share similar T-cell epitopes [52]. Several studies have attempted to further explore this school of thought, and their results have provided greater insight into this particular topic. Ad35 is one of the rarest human AdV serotypes, with a seroprevalence of less than 7% in the population. In an attempt to determine its potential as an alternative vector to Ad5, Barouch et al. compared the immunogenicity of a rAd35 vaccine expressing Simian Immunodeficiency Virus (SIV)-Gag versus a rAd5 SIV-Gag vaccine in Ad5 immune C57/BL6 mice [53]. Indeed, the efficacy of the rAd35-Gag vaccine was unaffected even by high levels of pre-existing Ad5 immunity, indicating the absence of a cross-reactive response between Ad5 and Ad35. Likewise, the presence of anti-Ad5 immunity did not affect the efficacy of a rAd11-Gag vector vaccine in mice. Interestingly, pre-existing anti-Ad11 immunity could suppress cellular immune responses elicited by the rAd35-Gag vaccine, suggesting some cross-reactivity between Ad11 and Ad35 immune responses [54]. The inhibitory effect of pre-existing antibodies to AdV remains a challenge for their use as viral vectors, and the development of alternative serotypes remains to be one of the best strategies to circumvent this limitation.

### 2.3. Influenza

The history of mankind is intricately intertwined with that of influenza, and the virus remains one of the top 10 threats to global health. Estimates indicate that there are approximately 1 billion cases globally each year, of which 3 to 5 million are severe cases and up to 650,000 eventually succumb to the disease [55]. Combined with the economic impact from the loss of work productivity, a proper and robust framework is required to counter this threat. The 2019 Global Influenza Strategies recommend vaccination as the most effective intervention to mitigate the impact of influenza [56]. Three types of influenza vaccines are licensed for use: (1) recombinant, (2) inactivated, and (3) live-attenuated. The three vaccines are multivalent and provide protection against selected influenza type A and type B strains that are predicted to spread in the upcoming season [57]. The former two are capable of inducing only IgG responses [57,58]. By contrast, the live-attenuated influenza vaccine (LAIV) can generate strain-specific IgG antibodies as well as mucosal IgA immunity and T-cell responses that are associated with protection from influenza illness [59,60]. Indeed, a meta-analysis by Ambrose et al. on the efficacy of LAIV in children showed that recipients of the live vaccine demonstrated a 44% reduction in influenza cases compared to those who received the trivalent inactivated vaccine (TIV) [61]. Put together, these studies demonstrate the potential of LAIV as a highly efficacious vaccine.

Yet, efforts to develop such a vaccine have been stymied by the presence of pre-existing immunity gained over a lifetime of exposure to different viral strains either from natural infection or vaccination. Indeed, several observational studies point to the possibility that pre-existing immunity can reduce the efficacy of both inactivated and life-attenuated influenza vaccine [62,63,64,65,66,67,68]. For example, Saito et al. found that children who received TIV for a previous season had reduced vaccine effectiveness for the current seasonal vaccine compared to unvaccinated children [66]. Likewise, Sasaki et al. showed lower antibody induction by a LAIV in individuals who had prior year TIV vaccination [67]. Furthermore, Coelingh et al. demonstrated that younger children aged two to eight as well as baseline seronegative adults had higher fold-induction of serum hemagglutinin inhibition (HAI) antibody titers post-LAIV vaccination [69]. However, given how complicated it is to trace the history of an individual’s exposure to influenza strains, it is difficult to tease out the exact impact that pre-existing immunity has on vaccine efficacy in human samples. Perhaps then, by establishing a good animal model with controlled infection histories, we will be able to better understand these complexities.

### 2.4. Flaviviruses

Flaviviruses include a number of clinically important pathogens that are either transmitted by mosquitoes (dengue, Zika, yellow fever, West Nile, and Japanese encephalitis virus) or by ticks (tick-borne encephalitis, Powassan virus) [70]. The recent Zika pandemic witnessed more than 3700 cases of congenital birth defects linked to Zika virus infection in the Americas and was declared by the World Health Organization in February 2016 to be a public health emergency [71]. Dengue infections, on the other hand, account for 390 million infections annually, of which 96 million infections are symptomatic [72]. The antigenic closeness between flaviviruses is such that infection with one flavivirus induces species-specific immunity, as well as cross-reactive antibodies against related serocomplexes [73,74]. However, these cross-reactive antibodies do not necessarily cross-protect. Initial insights come from human dengue challenge studies by Albert Sabin, who provided evidence that a homologous challenge in humans with the same dengue virus (DENV) serotype protects against re-infection, but only short-term protection against the heterologous DENV serotype challenge for up to six months [75,76]. Moreover, when cross-reactive antibodies decline to sub-neutralizing levels, these antibodies can opsonize dengue virus infection, resulting in enhanced virus burden and risk of severe dengue in patients experiencing secondary infection [77]. Indeed, recent cohort studies conducted in Nicaragua and Thailand provide clinical evidence that a specific range of antibody titers is associated with an increased risk of severe dengue [78,79]. The presence of waning maternal anti-dengue antibodies can also predispose children to dengue hemorrhagic fever, reinforcing the concept that pre-existing antibodies can promote disease pathogenesis [80].

Consistent with the notion that sub-neutralizing cross-reactive antibodies can promote viral infection, the presence of pre-existing cross-reactive antibodies can also increase the immunogenicity of flaviviral LAVs. As demonstrated in both Dengvaxia^®^ and Takaeda^®^ dengue vaccine trials, seropositive individuals produced greater neutralizing antibody responses and protection against the wild-type DENV infection compared to seronegative vaccinees [19,81,82]. These studies were also supported by in vivo studies showing that sequential immunization for flaviviruses with shared CD4 epitopes that could enhance protection during subsequent heterologous infection [83]. However, in another study, pre-existing antibodies from yellow fever vaccination can cause impairment of neutralizing antibody responses to tick-borne encephalitis vaccination [84]. The clinical trial finding that subjects with a limited range of cross-reactive antibodies from a prior Japanese Encephalitis vaccine were able to enhance yellow fever vaccination, by prolonging vaccine viremia duration that leads to higher antibody titers, thus hints at the possibility that whether pre-existing antibodies inhibit or augment flavivirus infection will depend on both antibody titers and the type/specificity of antibodies produced [85]. The plausible mechanisms involved are as elaborated below.

### 2.5. Interaction of Antibody–Virus Complexes with Immune Cells

The primary role of antibodies is antigen binding and interacting with Fc-gamma receptors (FcγRs) to modulate subsequent immune responses. The integration of both activating and inhibiting signals is critical for the generation of an effective immune response. In this aspect, FcγRs are an archetype of how such signals influence both innate and adaptive immune functions. Functionally, FcγRs can be classified into either activating or inhibiting receptors depending on the pathway they initiate [86]. Activating FcγRs possess an immunoreceptor tyrosine-based activation (ITAM) motif in their cytosolic domain, or in the case of FcγRI and FcγRIIIA, associate with an ITAM-containing γ-chain. Engagement of activating FcγRs by immune complexes results in the phosphorylation of the γ-chain by SRC-family kinases in order to create a docking site for spleen tyrosine kinase (SYK). Subsequent activation of SYK results in a signaling cascade, leading to the induction of pro-inflammatory responses and activation of innate immune effector cells [87,88]. By contrast, the inhibitory FcγRIIB receptor contains an immunoreceptor tyrosine-based inhibition (ITIM) motif within its intracellular domain. Cross-linking of FcγRIIB enables the recruitment of SH2 domain-containing inositol polyphosphate 5′ phosphatase (SHIP) and SH2 domain-containing protein tyrosine phosphatase 1 (SHP1) to modulate signals generated by activating FcγRs, thereby regulating the magnitude of inflammatory responses. Furthermore, FcγRIIB is important for controlling B-cell development [89,90,91,92,93,94]. Indeed, B-cells express FcγRIIB as the only FcγR on their cell surface, and cross-linking of FcγRIIB on naïve B-cells could inhibit their proliferation and differentiation into plasma cells [95]. Likewise, cross-linking of FcγRIIB induces apoptosis in bone marrow plasma cells, suggesting that FcγRIIB may influence the lifespan of these antibody-producing cells [89]. The signaling responses triggered by virus–antibody complexes are thus highly dependent on the type of FcγRs with which the immune complexes interact, which can result in either virus inhibition or enhancement of virus infection.

## 3. Mechanisms of Virus Inhibition Modulated by Pre-Existing Antibodies

### 3.1. Neutralization of Live-Attenuated Vaccines

Virus neutralization occurs when virions are bound by antibodies with stoichiometry exceeding a required threshold. Hence, one of the popular explanations to explain the lack of LAV efficacy in the presence of pre-existing antibodies is the neutralization of the LAV, which could consequently decrease the amount of viral antigens to levels that are below the threshold for immune detection and recognition. Antibody concentrations, affinity, and epitope accessibility are critical determinants for virus neutralization [96,97,98]. Antibody affinity, defined as the fraction of epitopes that are bound by antibodies at non-saturating concentrations, has been shown to correlate with neutralizing activity in vitro. On the other hand, epitope accessibility is defined as the number of epitopes on viruses that are available for binding and can be affected by virus structure, structural dynamics of virus, and virus maturation states [99]. The epitope availability would thus affect the fraction of epitope occupancy that will be required for virus neutralization. Taken together, cross-reactive antibodies that can neutralize virus infection are likely those that can bind to accessible epitopes with considerable affinity. Conversely, antibodies that bind weakly and target epitopes with reduced accessibility are unlikely to neutralize viruses, and may instead enhance viral infection via FcγR-mediated uptake.

Antibodies can neutralize LAV strains in a variety of ways, as summarized in Figure 1. They may block virus attachment and entry by either binding to epitopes that are directly involved in virus–receptor interactions or by imposing steric hindrance that prevent virus interaction with host receptors. As most virus structures are dynamic and can change structural conformations at different temperatures, it is thought that antibody binding to these dynamic structures may cause structural changes that can impair virus attachment, thereby causing virus neutralization [100]. However, the blockade of virus–receptor interactions alone may not be able to completely neutralize the viruses, especially in FcγR-bearing cells, as activating FcγRs can enable entry of virus–antibody complexes by FcγR-mediated uptake. Thus, pre-existing antibodies that can block viral fusion and uncoating will likely be more efficient in virus neutralization. In situations where pre-existing cross-reactive antibodies are unable to inhibit viral fusion processes intracellularly, high concentrations of antibodies may enable the formation of viral immune aggregates that influence the types of FcγRs engaged. These large viral aggregates can then inhibit phagocytosis by co-ligating the lowly expressed inhibitory receptor FcγRIIB that inhibit phagocytosis [101,102]. Finally, there have been theories suggesting that FcγR cross-linking by virus immune complexes may increase the production of IL-10 that abolishes innate immune responses [103]. However, more experimental evidence will be required to support this theory.

### 3.2. Inhibition of B-Cell Responses by Immune Complexes

Unlike myeloid cells, B-cells exclusively express FcγRIIB but not the activating FcγRs [86]. Therefore, cross-linking of the B-cell receptor (BCR) with FcγRIIB mediated by virus immune complexes can lead to the inhibition of B-cell activation (Figure 2a). Indeed, by adding sheep red blood cell-specific (SBRC) IgG to SRBCs, B-cell antibody secretion is reduced [104]. Similarly, using the cotton rat model of MV vaccination, maternal antibodies were demonstrated to inhibit B-cells by the cross-linking of BCR and FcγRIIB [105]. However, this mechanism of inhibition has not been demonstrated for other viruses. It is conceivable that this mode of inhibition is dependent on the size of the virus–antibody immune complexes, as immunization of small polypeptides can escape maternal antibody inhibition [106]. More mechanistic studies will hence be required to evaluate the conditions that need to be satisfied to cause B-cell inhibition.

Another way in which antibodies can inhibit B-cell responses is through epitope masking. This hypothesis postulates that the presence of pre-existing antibodies can mask the exposure of epitopes, thereby prohibiting recognition by the B-cell. This is also termed as epitope-specific suppression, whereby epitopes covered by these antibodies are unable to be recognized by B-cells (Figure 2b). Interestingly, there have been reports of epitope unspecific suppression, where monoclonal antibodies that target only one specific epitope can suppress B-cell recognition of a whole antigen, suggesting the possibility that steric hindrance or obstruction by high concentrations of antibodies can also lead to overall suppression B-cell recognition [107,108].

## 4. Mechanisms of Enhanced Vaccine Immunogenicity Modulated by Pre-Existing Antibodies

### 4.1. Antibody-Dependent Enhancement of LAV Infection and Immunogenicity

Antibody-dependent enhancement (ADE) of viral infection has been documented for many viruses, including flaviviruses, influenza, MV, Ross river viruses, HIV, and coronaviruses [109]. ADE can occur when sub-neutralizing levels of cross-reactive antibodies form immune complexes with viruses, opsonizing viral infection in FcγR-bearing myeloid cells including monocytes, macrophages, and dendritic cells via activating FcγR-mediated uptake. The majority of the mechanistic insights about ADE of viral infection are gathered from dengue, as waning cross-reactive antibodies that are acquired from a heterotypic DENV infection or through maternal-fetal transmission can result in a heightened risk of severe dengue that can be life-threatening [78,79,80]. Both activating FcγRI and FcγRIIA have been shown to be involved in ADE-mediated infection, although increasing studies indicate that FcγRIIA could be more important than FcγRI in enhancing viral infection [110,111]. While the precise mechanisms involved remain unclear, it is possible that the trafficking of immune complexes through FcγRIIA-mediated uptake is slower, thereby allowing more viral fusion and infection [112,113]. The activation of FcγRI, however, may aid to further enhance immunogenic responses to viral antigens by targeting virus immune complexes to the late endosomes or lysosomes for enhanced antigen processing and antigen presentation to the CD4+ T-cells, thereby increasing B-cell responses [112]. Overall, at sub-neutralizing antibody levels, the presence of pre-existing antibodies can activate both FcγRI and FcγRIIA, which promotes virus uptake, replication, and antigen presentation that consequently augments vaccine immunogenicity.

### 4.2. Intrinsic Host Responses That Promote Vaccine Immunogenicity

Besides promoting viral entry and antigen presentation, the cross-linking of FcγRs may also modulate cellular and host responses that promote viral replication and LAV immunogenicity (Figure 3). Some insights can be obtained from our clinical trial, where subjects were sequentially vaccinated with the inactivated Japanese Encephalitis virus vaccine followed by the yellow fever vaccine. Subjects within a restricted range of cross-reactive antibodies resulted in increased antibody responses, whereas too many or too few antibodies resulted in reduced antibody responses, indicating the possible role of ADE in augmented vaccine antibody responses [85]. In addition to the extended duration of viremia observed in these subjects, enhancing titers of cross-reactive antibodies provoked greater pro-inflammatory responses, including increased innate immune responses and the production of pro-inflammatory metabolites such as arachidonic acid, linoleic acid, and 12-HETE that promote phagocytosis and adaptive immune responses. The co-ligation of both FcγRI and FcγRIIA by virus–antibody immune complexes or cross-linking can also upregulate immune semaphorins such as SEMA4A, SEMA6A, and SEMA7A which are critical for antigen-presenting cell and T-cell interactions [85]. While the mechanisms of whether upregulation of immune semaphorins by immune complexes leads to increased T-cell proliferation and activation in humans remains to be evaluated, previous studies have shown that SEMA4A can enhance T-cell activation through interaction with Tim-2, thereby increasing antigen-specific T-cell and antibody responses against T-cell dependent antigens [114,115]. However, it is also noticeable that not all subjects within that specific window of cross-reactive antibody levels exhibited increased vaccine immunogenicity, suggesting that baseline variations, such as genetics, dietary or environmental factors may also influence the outcome of LAV immunogenicity [85]. Some recent studies hinted at the possibility that baseline variations in B-cell signatures and gene regulation could influence LAV reactogenicity and immunogenicity, which can be potential avenues for future studies [116,117].

## 5. Concluding Remarks

In this review, we have highlighted several studies that have shown that the efficacy of some commonly used LAVs, such as measles, adenovirus, influenza, and flaviviral vaccines, can be affected by these pre-existing adaptive immune responses. This knowledge will be critical to understanding the limitations of administering LAVs in seropositive individuals to reduce the incidence of vaccine failures, as well as in future designs of clinical trials to evaluate the efficacy of LAVs. Whether pre-existing antibodies can inhibit or augment LAV immunogenicity depends on the concentration and the type of antibodies that are present (Figure 4). With high antibody levels or potently neutralizing antibodies, pre-existing antibodies can inhibit LAV efficacy by virus neutralization or inhibition of B-cell responses. By contrast, at sub-neutralizing titers, pre-existing antibodies can enable viruses to better infect cells and increase innate immune responses that augment LAV immunogenicity. Moreover, pre-existing immunity can prime dendritic cells and memory T-cells to enhance protection during secondary infection with an antigenically related virus [83]. We believe that a deeper understanding of the underlying mechanisms involved will help us better understand the circumstances that can allow us to manage immunization in the presence of pre-existing antibodies, and even explore the possibilities of exploiting pre-existing antibodies to promote vaccine immunogenicity and efficacy. It would also be interesting to determine if cross-reactive antibodies can impact future development of LAVs against newly emerging pandemic viruses, including Ebola, Severe Acute Respiratory Syndrome coronavirus-2 (SARS CoV-2), and Zika viruses.

## Figures and Tables

**Figure 1 viruses-12-00520-f001:**
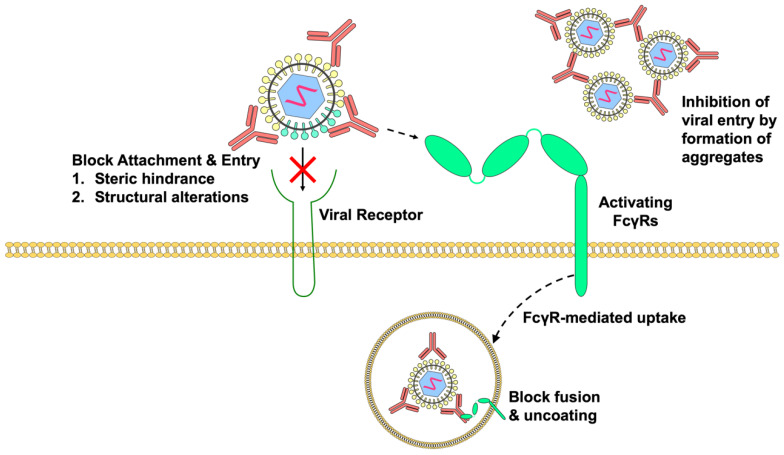
Mechanisms by which pre-existing antibodies neutralize LAVs. Antibodies can prevent attachment and entry of the virus, either via steric hindrance or by altering the conformation of the viral protein that binds the receptor. However, antibody binding to activating FcγRs may still allow virus–antibody complex internalization via FcγR-mediated uptake, indicating that antibodies that prevent viral fusion and uncoating could be critical to ensure complete virus neutralization. High levels of antibodies can also lead to the formation of aggregates which inhibit viral entry.

**Figure 2 viruses-12-00520-f002:**
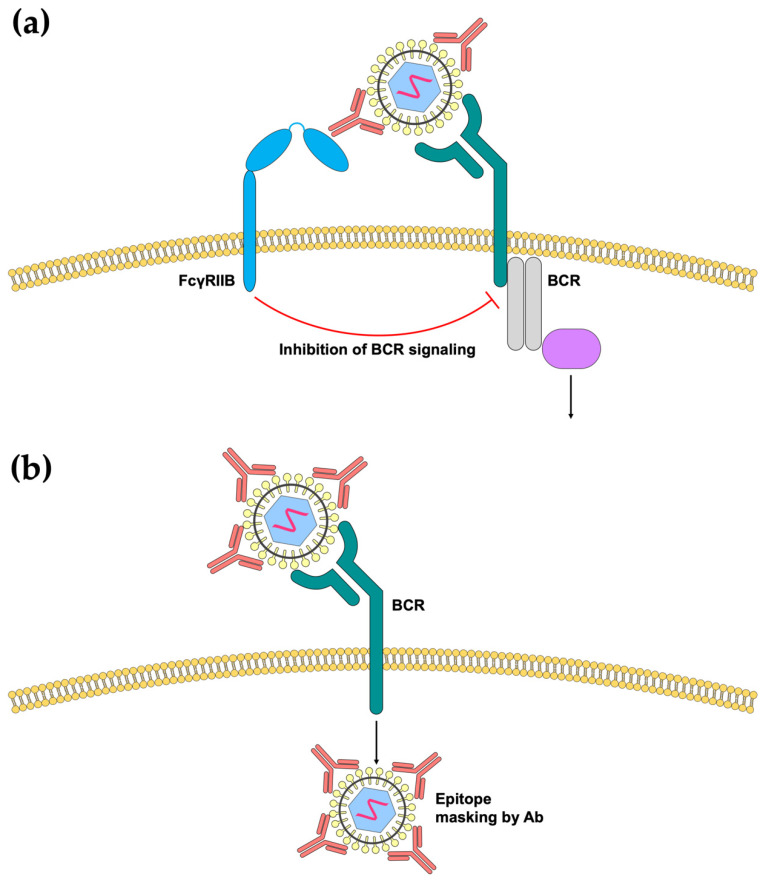
Effects of antibody binding on B-cell activation. (**a**) Formation of viral immune complexes may result in cross-linking and activation of the inhibitory FcγRIIB, which in turn inhibits downstream BCR signaling. (**b**) Antibody binding may prevent the exposure of epitopes normally recognized by B-cells in a phenomenon known as “epitope masking”, thereby preventing B-cell activation.

**Figure 3 viruses-12-00520-f003:**
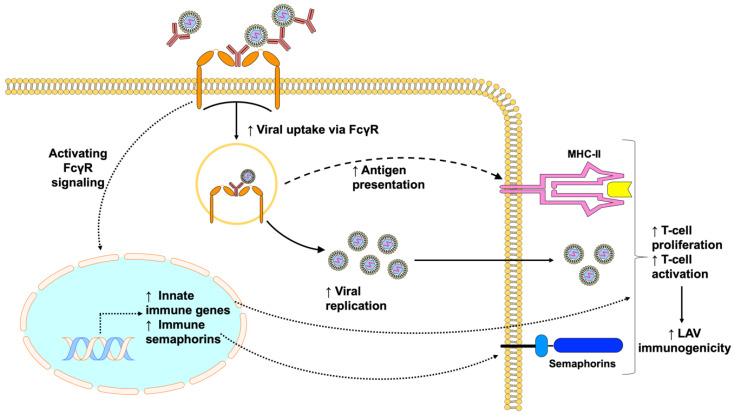
Pre-existing antibodies can improve vaccine immunogenicity by promoting virus infection and inducing innate and adaptive immune responses. Immune complexes formed by pre-existing antibodies and viruses can engage FcγRs, resulting in increased viral uptake and fusion through the process of antibody-dependent enhancement that leads to increased vaccine viremia and antigen presentation. Activating FcγR-signaling, on the other hand, provokes greater innate immune responses and production of pro-inflammatory metabolites that can enhance innate and adaptive immune responses. In addition, the cross-linking of FcγRs causes increased expression of immune semaphorins, which are critical for antigen-presenting cell and T-cell interaction. Overall, this leads to increased T-cell proliferation and activation, which consequently improves LAV immunogenicity.

**Figure 4 viruses-12-00520-f004:**
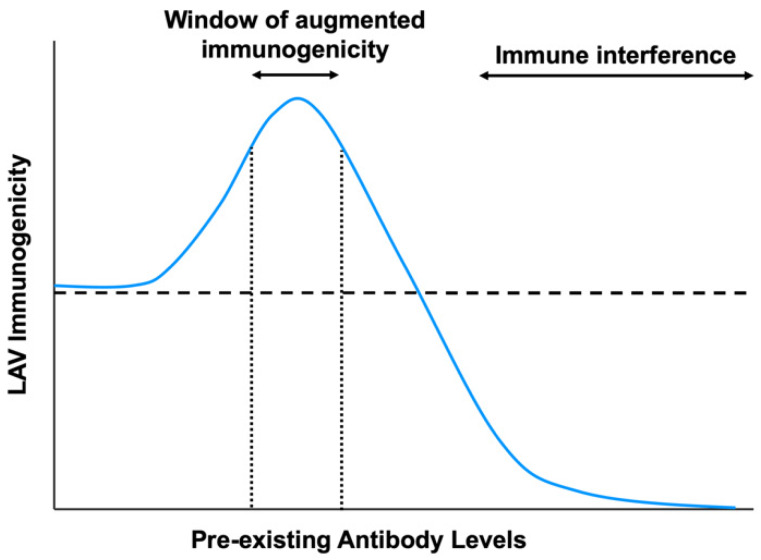
The impact of pre-existing antibody levels on live-attenuated vaccine immunogenicity. In the presence of high levels of pre-existing antibodies or with potently neutralizing antibodies, LAV immunogenicity is hindered due to immune interference from virus neutralization or inhibition of B-cell responses. However, at sub-neutralizing antibody titers, there is a window of augmented vaccine immunogenicity due to increased uptake into Fc receptor-bearing cells and greater induction of innate immune responses. The horizontal dotted line indicates LAV immunogenicity in the absence of pre-existing antibodies.

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
