# Peer review of "The Effects of Pre-Existing Antibodies on Live-Attenuated Viral Vaccines"

_viruses, 2020, doi:10.3390/v12050520_

Round 1

Reviewer 1 Report

The manuscript entitled ¨The effects of pre-existing antibodies on live-2 attenuated viral vaccines¨ by Darren Z. L. Mok and Kuan Rong Chan highlights several studies that have shown that the efficacy of some commonly used LAVs such as measles, adenovirus, influenza and flaviviral vaccines can be affected by pre-existing adaptive immune responses. This review examines the literature on this topic, and examines the circumstances where pre-existing antibodies can inhibit or enhance LAV efficacy. The figures facilitate the understanding of the text and the bibliography is adequate and updated. As a caveat, the authors write COVID-19 virus on line 425. COVID-19 is the name of the disease but the name of the virus is SARS-CoV2.

Author Response

Response to Reviewer 1 Comments

  1. As a caveat, the authors write COVID-19 virus on line 425. COVID-19 is the name of the disease but the name of the virus is SARS-CoV2.

We thank Reviewer 1 for the feedback and have changed the name from COVID-19 to Severe Acute Respiratory Syndrome coronavirus-2 (SARS CoV-2) (Line 429).

Reviewer 2 Report

This is a very good review in vaccine field although the authors only discussed the LAV (live attenuated virus).

The authors need to go over the manuscript to correct some typos. no other comments.

Author Response

The authors need to go over the manuscript to correct some typos. no other comments.

We thank Reviewer 2 for the comments and have systematically looked through the manuscript to correct the typos.

Reviewer 3 Report

The manuscript entitled "The effects of pre-existing antibodies on live-attenuated viral vaccines" is a well-written, comprehensive review of the literature regarding LAV immunogenicity. The authors provide a concise summary of the available literature and describe in some detail the mechanisms of immune alteration caused by pre-existing antibodies. There are some minor language and spelling errors, including several tense-discrepancies ("has" vs "have" line 216 and 351). Otherwise, English-language and text editing are minimal. 

References are needed at the end of line 50 and line 76.

Line 129 doesn't read well, should consider re-phrasing as, "Some of the first evidence..."

Line 237 states that "All flaviviruses are antigenically related and can, hence, induce broad cross-reactive antibodies that can bind broadly with other flaviviruses" is missing a reference and may not be true. For example, is there any evidence that prior infection with Hepatitis E virus cross-reacts with Dengue? Consider fact checking and referencing this statement.

In the concluding remarks, the authors fail to mention that while cross-reactive antibodies may lessen the response to LAV vaccination, they may also be quite protective against certain infections and don't always lead to ADE. A more comprehensive statement about the potential benefit of pre-existing immunity should be included int he concluding remarks.

This reviewer does not understand the value of Figure 4. Figure 4 makes it seem as though LAV's are either going to cause ADE or not work at all. consider either removing this figure or changing the dotted line in the middle to suggest that there is actually a window in which LAVs are highly effective, not a single point in time. 

Author Response

Response to Reviewer 3 comments

1. There are some minor language and spelling errors, including several tense-discrepancies ("has" vs "have" line 216 and 351). Otherwise, English-language and text editing are minimal. 

We thank the Reviewer for pointing out the tense-discrepancies that we have overlooked and have changed the tenses from “has” to “have” (now at lines 242 and 402).

2. References are needed at the end of line 50 and line 76.

The appropriate references have been added (Ref. 11, 16, and 17).

3. Line 129 doesn't read well, should consider re-phrasing as, "Some of the first evidence..."

We thank Reviewer 3 for the recommendation and have re-phrased the sentence as suggested for better clarity (line 141).

4. Line 237 states that "All flaviviruses are antigenically related and can, hence, induce broad cross-reactive antibodies that can bind broadly with other flaviviruses" is missing a reference and may not be true. For example, is there any evidence that prior infection with Hepatitis E virus cross-reacts with Dengue? Consider fact checking and referencing this statement.

We thank the reviewer for pointing out that not all flaviviruses may produce antibodies that cross-react with other distantly-related flaviviruses. Indeed, as the reviewer pointed out, Hepatitis C virus has not been shown to cross-react with flaviviruses like dengue. We have hence updated our text to reflect that only antibodies that target related serocomplexes can cross-react (lines 263-265), and addionally, add in the relevant citations.

5. In the concluding remarks, the authors fail to mention that while cross-reactive antibodies may lessen the response to LAV vaccination, they may also be quite protective against certain infections and don't always lead to ADE. A more comprehensive statement about the potential benefit of pre-existing immunity should be included in the concluding remarks.

We agree with the Reviewer that pre-existing immunity can be protective. Indeed, a previous study elegantly demonstrated that pre-existing Japanese Encephalitis Virus (JEV) immunity protects against a secondary dengue virus infection. As recommended by the reviewer, we have now added this information in the manuscript (lines 476-477).

6. This reviewer does not understand the value of Figure 4. Figure 4 makes it seem as though LAV's are either going to cause ADE or not work at all. Consider either removing this figure or changing the dotted line in the middle to suggest that there is actually a window in which LAVs are highly effective, not a single point in time.

We agree with Reviewer 3 that ADE of LAV may not be the most appropriate term to describe the effects of augmented immunogenicity. We think that the reviewer has provided a good suggestion, where we can describe this range as “a window of augmented vaccine immunogenicity”. According to the reviewer’s suggestion, we have now edited Figure 4 and the figure legend to provide greater clarity.